# Analyzing scRNA-seq data by CCP-assisted UMAP and tSNE

Yuta Hozumi [1¤], Guo-Wei Wei [1,2,3]*

**1** Department of Mathematics, Michigan State University, East Lansing, Michigan, United States of America, **2** Department of Electrical and Computer Engineering, Michigan State University, East Lansing, Michigan, United States of America, **3** Department of Biochemistry and Molecular Biology, Michigan State University, East Lansing, Michigan, United States of America

¤ Current address: School of Mathematics, Georgia Institute of Technology, Atlanta, Georgia, United States of America

* weig@msu.edu

## Abstract

Single-cell RNA sequencing (scRNA-seq) is widely used to reveal heterogeneity in cells, which has given us insights into cell-cell communication, cell differentiation, and differential gene expression. However, analyzing scRNA-seq data is a challenge due to sparsity and the large number of genes involved. Therefore, dimensionality reduction and feature selection are important for removing spurious signals and enhancing downstream analysis. Correlated clustering and projection (CCP) was recently introduced as an effective method for preprocessing scRNA-seq data. CCP utilizes gene-gene correlations to partition the genes and, based on the partition, employs cell-cell interactions to obtain super-genes. Because CCP is a data-domain approach that does not require matrix diagonalization, it can be used in many downstream machine learning tasks. In this work, we utilize CCP as an initialization tool for uniform manifold approximation and projection (UMAP) and t-distributed stochastic neighbor embedding (tSNE). By using 21 publicly available datasets, we have found that CCP significantly improves UMAP and tSNE visualization and dramatically improve their accuracy. More specifically, CCP improves UMAP by 22% in ARI, 14% in NMI and 15% in ECM, and improves tSNE by 11% in ARI, 9% in NMI and 8% in ECM.

**Data Availability Statement:** All data is publicly available from the Gene Expression Omnibus database with the accession numbers GSE75748, GSE82187, GSE94820, GSE67835, GSE84133, GSE109774, GSE85241, GSE74672, SCP1749 We have uploaded the data and code to reproduce our

## Introduction

Single-cell RNA sequencing (scRNA-seq) is a relatively new technology that profiles the transcriptome of individual cells within a tissue or organ, aiming to gain understanding of gene expression, gene regulation, cell-cell interaction, spatial transcriptomics, signal transduction pathways, and more [1]. The typical workflow of scRNA-seq involves cell isolation, RNA extraction, library preparation, sequencing, and data analysis. Through technological advances in the experimental procedures, the read quality has improved, and over 10,000 samples can now be sequenced at once. Despite the improvements, the data still contains nonuniform noise, is often times unlabeled and has high dimensions. In order to analyze such complex data, a standard data analysis pipeline involves data preprocessing, gene expression

work in figshare, under https://doi.org/10.6084/m9.figshare.26501389.v7.

**Funding:** National Institute of health (NIH) grants R01GM126189, R01AI164266, and R35GM148196. National Science Foundation (NSF) grants DMS-2052983, DMS-1761320, and IIS-1900473 National Aeronautics and Space Administration (NASA) grant 80NSSC21M0023 Michigan State University (MSU) Foundation Bristol-Myers Squibb 65109 Pfizer. The funders had no role in study design, data collection and analysis, decision to publish, or preparation of the manuscript.

**Competing interests:** The authors have declared that no competing interests exist.

quantification, normalization and batch correction, dimensionality reduction, cell type identification, differential gene expression analysis, and pathway and functional analysis [2–7]. An effective dimensionality reduction must be employed in order to have a meaningful downstream analysis.

Two of the most popular dimensionality reductions for scRNA-seq are principal components analysis (PCA) and nonnegative matrix factorization (NMF). The first component is called the principal component, where the variance of the projected data is maximized. The subsequent $i$th component is orthogonal to the first $i-1$ components, and maximizes the variance of the residual data projected onto the $i$th component [8]. PCA aims to obtain a lower dimensional representation of the data to identify important gene patterns. Many PCA derivatives have also been used for scRNA-seq data analysis [9–12]. In particular, a popular package Seurat [13] utilizes supervised PCA to find an optimal projection that incorporate local structure of the reference data for the downstream analysis. However, because PCA requires matrix diagnolization and its projected data contains negative values, it is difficult to interpret. In contrast, NMF has an additional constrain such that the low dimensional representation is nonnegative. Each components, often called a metagene in scRNA-seq analysis, is a linear combination of original genes [14]. NMF has seen a numerous extension to the original formulation, including robustness to noise and manifold regularization [15–21]. Through the nonnegative constrain, NMF is highly interpretable, and may be more suitable downstream analysis.

Deep learning and ensemble methods are another class of approaches that have become popular for single cell RNA-seq analysis. Single-cell variational inference (scVI) utilizes deep neural networks to obtain information from similar cells and genes to approximate the distribution of underlying gene expression values [22]. Single-cell cluster using marker genes (SCMcluster) utilizes known marker genes to guide feature selection and perform ensemble clustering [23]. AutoCell [24] utilizes variational autoencoding network that combines the Gaussian mixture model and graph embedding to model the high dimensional scRNA-seq data. Diffusion models [25–28], generative adversarial network (GAN) [27], language models [29–32], transformers [33–37], ensemble methods [38–40] and more [41–44] have also been used for scRNA-seq analysis. Though these methods have great performance, they rely on careful curation of data and often require large amount of data for pretraining.

Visualization of the data is also an important aspect of scRNA-seq analysis pipeline. After data preprocessing and feature extraction through dimensionality reduction, the visualization of data commonly involves the utilization of uniform manifold and projection (UMAP) or t-distributed stochastic neighbor embedding (tSNE) [45, 46]. UMAP obtains its visualization by constructing a $k$-dimensional weighted graph and computes the edge-wise cross entropy between the weighted graph of the low dimensional embedding and the $k$-dimensional weighted graph of the original space. Through the utilization of stochastic gradient descent, UMAP demonstrates a notable improvement in speed and scalability compared to other Laplacian eigenmap-based algorithms. TSNE computes data similarity by constructing a conditional probability distribution among pairs of data points. It employs the Student's t-distribution to derive the probability distribution of the low-dimensional embedded space. Subsequently, it minimizes the Kullback-Leibler (KL) divergence between these two distributions to obtain the visualization [46, 47]. These visualization methods are widely utilized in popular analysis pipelines such as Scanpy for Python and Seurat for R. They serve as crucial tools to ensure proper feature selection and facilitate comprehensive data exploration.

We proposed correlated clustering and projection (CCP) as a general approach for dimensionality reduction [48, 49]. CCP is a data-domain method that completely bypasses matrix diagonalization. It partitions genes into clusters based on their similarities and then

projects genes in the same clusters into a super-gene, which is a measure of accumulated gene-gene correlations among cells. The gene partition can be realized by either the standard $k$-means or the $k$-medoids using either covariance distance or correlation distance. Flexibility rigidity index (FRI) is used for the nonlinear projection [50]. The resulting super-genes are all non-negative and highly interpretable. Recently, CCP has been applied to the clustering and classification of scRNA-seq datasets [49], where it showed an improvement over PCA across 14 scRNA-seq data. This indicates CCP's promising performance in handling single-cell RNA sequencing data.

The aim of this study is to explore CCP's potential as the primary dimensionality reduction method for visualization, particularly focusing on its application in initializing UMAP and tSNE, two highly effective visualization tools in scRNA-seq analysis. Additionally, we introduce a novel method for handling low-variance (LV) genes. Instead of discarding low-variance genes like many other methods, we group them together into a single category. This grouping is achieved by projecting them into one descriptor using FRI. One of the drawbacks of dropping low-variance genes is that scRNA-seq data often has an unequal number of cell types. Moreover, there are numerous genes with low expression, and removing too many genes may result in overlooking cell outliers. Therefore, LV-gene addresses this issue by consolidating low-variance genes into one descriptor, thereby increasing its predictive power. Through experimentation on 21 publicly available datasets, we evaluated CCP-assisted UMAP and CCP-assisted tSNE. Our findings showcase that CCP enhances UMAP by 22% in Adjusted Rand Index (ARI), 14% in Normalized Mutual Information (NMI), and 15% in Element-Centric Measure (ECM). Similarly, CCP improves tSNE by 11% in ARI, 9% in NMI, and 8% in ECM.

## Methods and algorithms

Consider a scRNA-seq dataset $\mathcal{Z} := \{z_m^i\}_{m=1,i=1}^{M,I}$, where $M$ is the number of cells and $I$ is the number of genes. CCP finds an $N$-dimensional representation $\mathcal{X} := \{x_m^n\}_{m=1,n=1}^{M,N}$, in which $1 \leq N << I$, by using a data-domain two-step strategy: gene clustering and gene projection. Fig 1 shows the workflow of CCP, and the details of each step is outlined below. First, the genes are clustered according to their similarities. Then, for each gene cluster, the genes are projected into 1 descriptor called the super-gene. The resulting component $x_m^n$ can be regarded as the $n$th super-gene for the $m$th cell. Then, subsequence analysis, such as 2D visualization using UMAP and tSNE can be performed.

### Gene clustering

To facilitate a gene clustering, we emphasize gene vectors by setting the original data as $\mathcal{Z} = \{\mathbf{z}^1, ..., \mathbf{z}^i, ..., \mathbf{z}^I\}$, where $\mathbf{z}^i \in \mathbb{R}^M$ represents the $i$th gene vector for the data. CCP partitions the gene vector into $N$ components with $1 \leq N << I$ by a clustering technique, such as $k$-means or $k$-medoids. CCP seeks an optimal disjoint partition of the data $\mathcal{Z} := \uplus_{n=1}^N \mathcal{Z}^n$, for a given $N$, where $\mathcal{Z}^n$ is the $n$th partition (cluster) of the genes. To this end, the correlations among gene vectors $\mathbf{z}^i$ are analyzed according to appropriate correlation measures, such as covariance distance and correlation distance. Note that the clustering is performed on the genes, rather than the cells.

Let $S = \{1, ..., I\}$ be the enumeration of the genes. We can partition $S$ into $S^1, ..., S^N$ using the gene clustering results by letting $S^n = \{i | \mathbf{z}^i \in \mathcal{Z}^n\}$. Then, $\mathbf{z}_m^{S^n}$ denotes the $S^n$ genes of the $m$th cell. Further detail can be found in Section S1.1 of the S1 File.

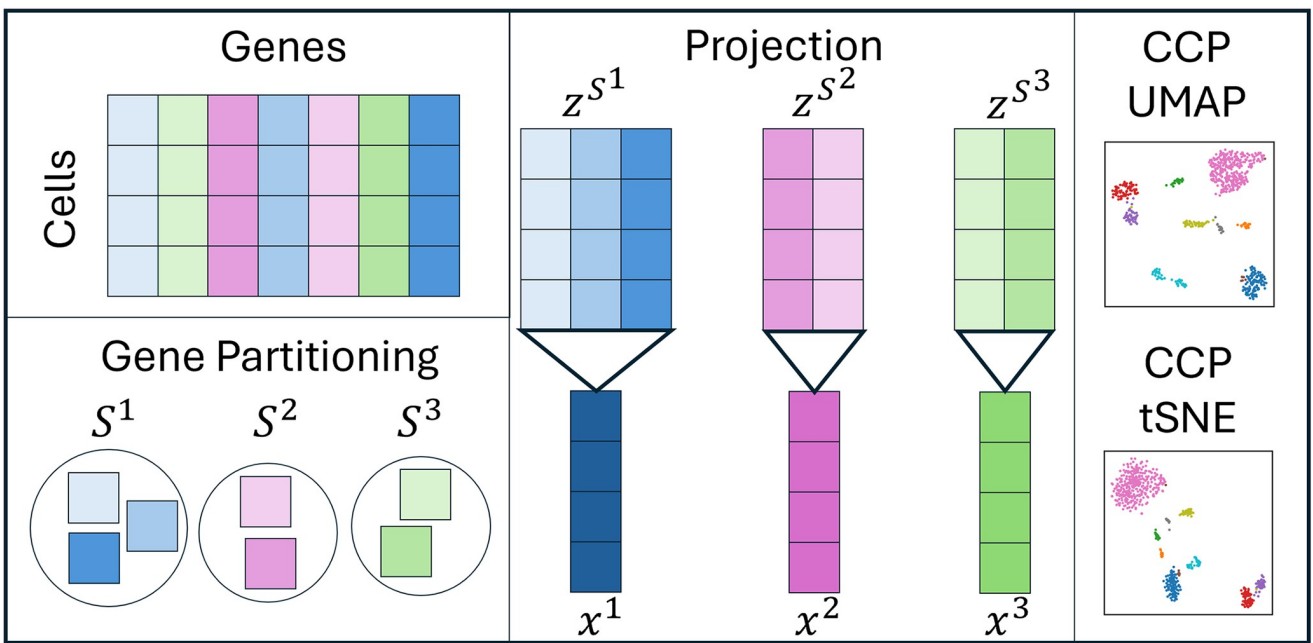

**Fig 1. Workflow of CCP.** From the input gene expression matrix, the genes are partitioned into groups, according to their similarity. Then, each group of genes is projected into 1 descriptor called super-gene. UMAP and tSNE can further be applied to the super-gene to visualize the cells in 2 dimensions.

**Gene projection.** Based on the gene partitioning, we denote $\mathbf{z}_m^{S^n} \in \mathbb{R}^{S^n}$ as $n$th cluster of $S^n$ genes for the $m$th cell. CCP projects these genes into a super-gene $x_m^n$ by using the flexibility rigidity index (FRI). Denote $\|\mathbf{z}_i^{S^n} - \mathbf{z}_j^{S^n}\|$ as some metric between cell $i$ and cell $j$ for the $n$th cluster of $S^n$ genes. The gene-gene correlation between the two cells are defined by $C_{ij}^{S^n} = \Phi(\|\mathbf{z}_i^{S^n} - \mathbf{z}_j^{S^n}\|; \eta^{S^n}, \tau, \kappa)$, where $\Phi$ is a correlation kernel, with parameters $\eta^{S^n}, \tau$, and $\kappa > 0$. One may use the Euclidean, Manhattan, and/or Wasserstein distances to measure the correlations. Additionally, the FRI correlation kernels satisfy the following conditions

$$\Phi(\|\mathbf{z}_i^{S^n} - \mathbf{z}_j^{S^n}\|; \eta^{S^n}, \tau, \kappa) \to 0, \quad \text{as} \|\mathbf{z}_i^{S^n} - \mathbf{z}_j^{S^n}\| \to \infty \tag{1}$$

$$\Phi(\|\mathbf{z}_i^{S^n} - \mathbf{z}_j^{S^n}\|; \eta^{S^n}, \tau, \kappa) \to 1, \quad \text{as} \|\mathbf{z}_i^{S^n} - \mathbf{z}_j^{S^n}\| \to 0. \tag{2}$$

Although various radial basis functions can be used in CCP, we consider generalized exponential function in the present work

$$\Phi(\|\mathbf{z}_i^{S^n} - \mathbf{z}_j^{S^n}\|; \eta^{S^n}, \tau, \kappa) = \begin{cases} e^{-\left(\frac{\|\mathbf{z}_i^{S^n} - \mathbf{z}_j^{S^n}\|}{\eta^{S^n}\tau}\right)^{\kappa}} & \|\mathbf{z}_i^{S^n} - \mathbf{z}_j^{S^n}\| < r_c^{S^n} \\ 0, & \text{otherwise.} \end{cases} \tag{3}$$

where $r_c^{S^n}$ is the cutoff distance and $\eta^{S^n}$ is the scale, which are defined from the data automatically. Here, $\kappa$ is the power and $\tau$ is a scale parameter.

The gene-gene correlation matrix $C^{S^n} = \{C_{ij}^{S^n}\}$ represents the cell-cell interactions for genes $S^n$, and it captures all the interaction up to a threshold, which is determined by $r_c^{S^n}$. Here, we take $r_c^{S^n}$ to be the 3-standard deviations of the pairwise distances. Additionally, to automatically

evaluate $\eta^{S^n}$, we consider the average minimal distance between the cluster of genes

$$\eta^{S^n} = \frac{\sum_{m=1}^{M} \min_{\mathbf{z}_j^{S^n}} \|\mathbf{z}_m^{S^n} - \mathbf{z}_j^{S^n}\|}{M}. \tag{4}$$

Using the correlation function, CCP projects $S^n$ genes into a super-gene using FRI for $i$th sample,

$$x_i^n = \sum_{m=1}^{M} w_{im} \Phi(\|\mathbf{z}_i^{S^n} - \mathbf{z}_m^{S^n}\|; \eta^{S^n}, \tau, \kappa), \tag{5}$$

where $w_{im}$ are the weights. In this work, we set $\omega_{im} = 1$.

CCP obtains the lower dimensional super-gene representation for $i$th sample (cell) $\mathbf{x}_i = (x_i^1, ..., x_i^N)^T$ by running the projection for all gene clusters $\{\mathcal{Z}^n\}$.

**Low variance (LV) genes.** One major challenge of scRNA-seq analysis is dealing with sparsity and low variance genes. We propose using low variance (LV) genes to collapse the low varying genes into 1 super-genes to increase their predictive power.

Let $\mathbf{v} = (v_1, \ldots, v_I)$ be the variance of the genes, where $v_i$ is the variance of gene $\mathbf{z}^i$, and assume that the variance are sorted in descending order. Then, define the low variance set $P$ as

$$P = \{i | i > v_c I\}$$

where $0 \leq v_c \leq 1$ is the cutoff ratio. Then, we can obtain the cell-cell correlation using these low variance genes $C_{ij}^P$,

$$C_{ij}^P = \Phi(\|\mathbf{z}_i^P - \mathbf{z}_j^P\|; \eta^P, \tau, \kappa)$$

where $\Phi(\|\mathbf{z}_i^P - \mathbf{z}_j^P\|; \eta^P, \tau, \kappa)$ is the generalized exponential function

$$\Phi(\|\mathbf{z}_i^P - \mathbf{z}_j^P\|; \eta^P, \tau, \kappa) = \begin{cases} e^{-\left(\frac{\|\mathbf{z}_i^P - \mathbf{z}_j^P\|}{\eta^P \tau}\right)^{\kappa}} & \|\mathbf{z}_i^P - \mathbf{z}_j^P\| < r_c^P \\ 0, & \text{otherwise.} \end{cases}$$

$r_c^P$ is taken as the 3-standard deviation of the pairwise distances, and $\eta^P$ is the average minimum distance

$$\eta^P = \frac{\sum_{m=1}^{M} \min_{\mathbf{z}_j^P} \|\mathbf{z}_m^P - \mathbf{z}_j^P\|}{M}.$$

Using the correlation function, CCP projects $|P|$ genes into a super-gene using FRI for $i$th sample,

$$x_i = \sum_{m=1}^{M} w_{im} \Phi(\|\mathbf{z}_i^P - \mathbf{z}_m^P\|; \eta^{S^n}, \tau, \kappa),$$

where $w_{im}$ are the weights.

For CCP, we compute the LV-gene first, and use the correlated partition algorithm on the remaining genes.

# Results

## Data preprocessing

We have tested CCP-assisted UMAP and tSNE visualization on 21 publicly available data. Table 1 displays information including the Gene Expression Omnibus (GEO) accession ID [51, 52], the reference, data dimensions, and cell composition for each dataset. Additionally, data from the scziDesk paper [53] was utilized and can be accessed from their S1 File. The Qx and Qs data correspond to Smart-seq2 and 10x genomic data from Quake et al. [54]. Notably, the GSE84133 human dataset encompasses all human patient data from Baron et al. [55]. Detailed statistics for each data can be found in S1 Table in the S1 File.

To normalize the data, we began by normalizing the counts by using the average median gene count of each cell. Let $X \in \mathbb{R}^{M \times N}$ be the data, with $M$ cells and $N$ genes. Each row (cell) was divided by its row sum, followed by multiplication by the median row sum to obtain a normalized count matrix. Finally, log-transform was applied using Scanpy's log1p method.

In our benchmarking process, we employed CCP with parameters $\tau = 6$ and $\kappa = 2$ to reduce the dimensions to 300 super-genes. Additionally, we utilized $\nu = 0.8$ to generate the LV-gene. Clustering was performed using the Leiden algorithm, and we evaluated the quality of clustering using ARI, NMI and ECM by comparing the obtained clusters with the cell types provided by the original authors. Visualizations were generated using Scanpy's implementation of UMAP and tSNE. In order to reduce the computation load for datasets exceeding 2,000 samples, we utilized subsampling. For all the benchmarking, we utilized Michigan State University's high performance computing cluster, which utilizes AMD EPYC 7H12 Processor. We utilized 16gb of memory with 4 CPU cores.

**Table 1. Dataset name, reference, dimensions and cell type composition.**

| Dataset [Ref] | Size (cells x genes) | Cell Composition |
|---|---|---|
| GSE75748cell [56] | 1018 x 19097 | 7 clusters: 138, 105, 212, 162, 159, 173, 69 |
| GSE75748time [56] | 758 x 19189 | 6 clusters: 92, 102, 66, 172, 138, 188 |
| GSE82187 [57] | 705 x 18840 | 10 clusters: 107, 18, 21, 71, 48, 7, 334, 13, 43, 43 |
| GSE67835 [58] | 420 x 22084 | 8 clusters: 18, 62, 20, 110, 25, 16, 131, 38 |
| GSE84133 H1 [55] | 1937 x 20125 | 14 clusters: 110, 51, 236, 872, 214, 120, 130, 13, 70, 14, 8, 92, 5, 2 |
| GSE84133 H2 [55] | 1724 x 20125 | 14 clusters: 3, 81, 676, 371, 125, 301, 23, 2, 86, 17, 9, 22, 6, 2 |
| GSE84133 H3 [55] | 3605 x 20125 | 14 clusters: 843, 100, 1130, 787, 161, 376, 92, 2, 36, 14, 7, 54, 1, 2 |
| GSE84133 H4 [55] | 1303 x 20125 | 14 clusters: 2, 52, 284, 495, 101, 280, 7, 1, 63, 10, 1, 5, 1, 1 |
| GSE84133 M1 [55] | 822 x 14878 | 13 clusters: 2, 4, 4, 9, 343, 85, 236, 72, 14, 4, 17, 29, 3 |
| GSE84133 M2 [55] | 1064 x 14878 | 13 clusters: 8, 3, 10, 182, 551, 133, 39, 67, 27, 4, 19, 18, 3 |
| GSE84133 human [55] | 8569 x 20125 | 14 clusters: 958, 284, 2326, 2525, 601, 1077, 252, 18, 255, 55, 25, 173, 13, 7 |
| Muraro [59] | 2122 x 19046 | 9 clusters: 21, 812, 193, 101, 219, 245, 3, 80, 448 |
| Romanov [60] | 2881 x 21143 | 7 clusters: 267, 240, 356, 48, 898, 1001, 71 |
| Qx Bladder [54] | 2500 x 23341 | 4 clusters: 1203, 1167, 57, 73 |
| Qx Limb Muscle [54] | 3909 x 23341 | 6 clusters: 461, 320, 1330, 308, 1136, 354 |
| Qx Spleen [54] | 9552 x 23341 | 5 clusters: 6886, 1930, 42, 464, 230 |
| Qs Diaphragm [54] | 870 x 23341 | 5 clusters: 78, 81, 31, 241, 439 |
| Qs Limb Muscle [54] | 1090 x 23341 | 6 clusters: 71, 35, 141, 45, 258, 540 |
| Qs Lung [54] | 1676 x 23341 | 11 clusters: 57, 53, 25, 90, 113, 35, 693, 65, 85, 37, 423 |
| Qs Trachea [54] | 1350 x 23341 | 4 clusters: 206, 113, 201, 830 |
| SCP1749 [61] | 10006 x 24820 | 11 clusters: 437, 206, 139, 122, 5430, 364, 2713, 90, 44, 94, 367 |

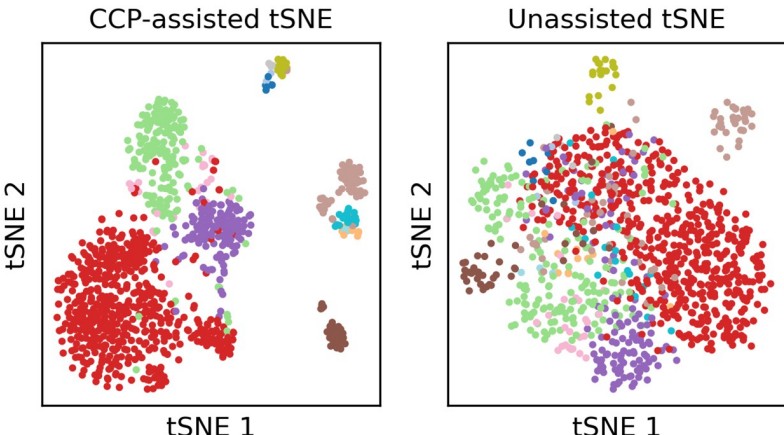

**Fig 2. TSNE visualization of GSE84133 mouse2 data.** The left and right figures show the CCP-assisted and the standard tSNE visualization.

## Visualization

Preprocessing of scRNA-seq data is a key step for visualization. Fig 2 shows an example of CCP-assisted tSNE visualization and the original tSNE visualization of the Baron dataset [55]. The original data has 20,125 genes, and aggressively reducing the original dimension to 2 dimensions by tSNE leads to poor visualization. In CCP-assisted tSNE, CCP was utilized to reduce the original genes into 300 super-genes, which were further reduced to 2 dimensions with tSNE for visualization. It is clear that CCP-assisted tSNE significantly improves the visualization quality in this case. We further showcase CCP-assisted visualization on the dataset described in Table 1. We provide additional comparison with PCA-assisted and NMF-assisted visualization in Section S2.2 of the S1 File.

Fig 3 show the comparison of CCP-assisted UMAP and tSNE with standard UMAP and tSNE visualization on Quake dataset. Each row correspond to one of the 5 dataset, and the columns correspond to CCP-assisted UMAP, CCP-assisted tSNE, standard UMAP and standard tSNE visualization. The samples were colored according to the true cell type.

CCP improves the overall visualization of the Quake dataset. In Qx Bladder data, CCP-assisted UMAP and tSNE show 1 bladder cluster, whereas the standard UMAP and tSNE visualization show an elongated cluster of bladder cells. The urotherial cells in CCP-assisted UMAP and tSNE are divided into 3 subclusters, whereas the standard UMAP and tSNE visualization show 1 cluster. In Qs Diaphragm data, CCP-assisted UMAP and tSNE show 5 distinct clusters corresponding to each cell types. However the UMAP visualization do not differentiate the 5 cell types. The standard tSNE visualization show poor clustering, where satellite cell, mesenchymal cell and endothelial cell form a supercluster. In Qs Limb Muscle cell, all visualization show a supercluster of B cell and T cell. CCP-assisted visualization show a clear distinction between the B-T cell supercluster and macrophages, whereas the standard visualization show a supercluster of B cell, T cell, macrophages and endothelial cells. In the Qs Trachea data, the standard UMAP and tSNE visualization show a subpopulation of mesenchymal cell within the epithelial cell, whereas CCP-assisted counterparts do not.

Fig 4 show the comparison of CCP-assisted UMAP and tSNE with standard UMAP and tSNE visualization on GSE75748 cell, GSE75748 time, GSE67835 and GSE82187 dataset. The columns correspond to CCP-assisted UMAP, CCP-assisted tSNE, standard UMAP and standard tSNE visualization. The samples were colored according to the true cell type.

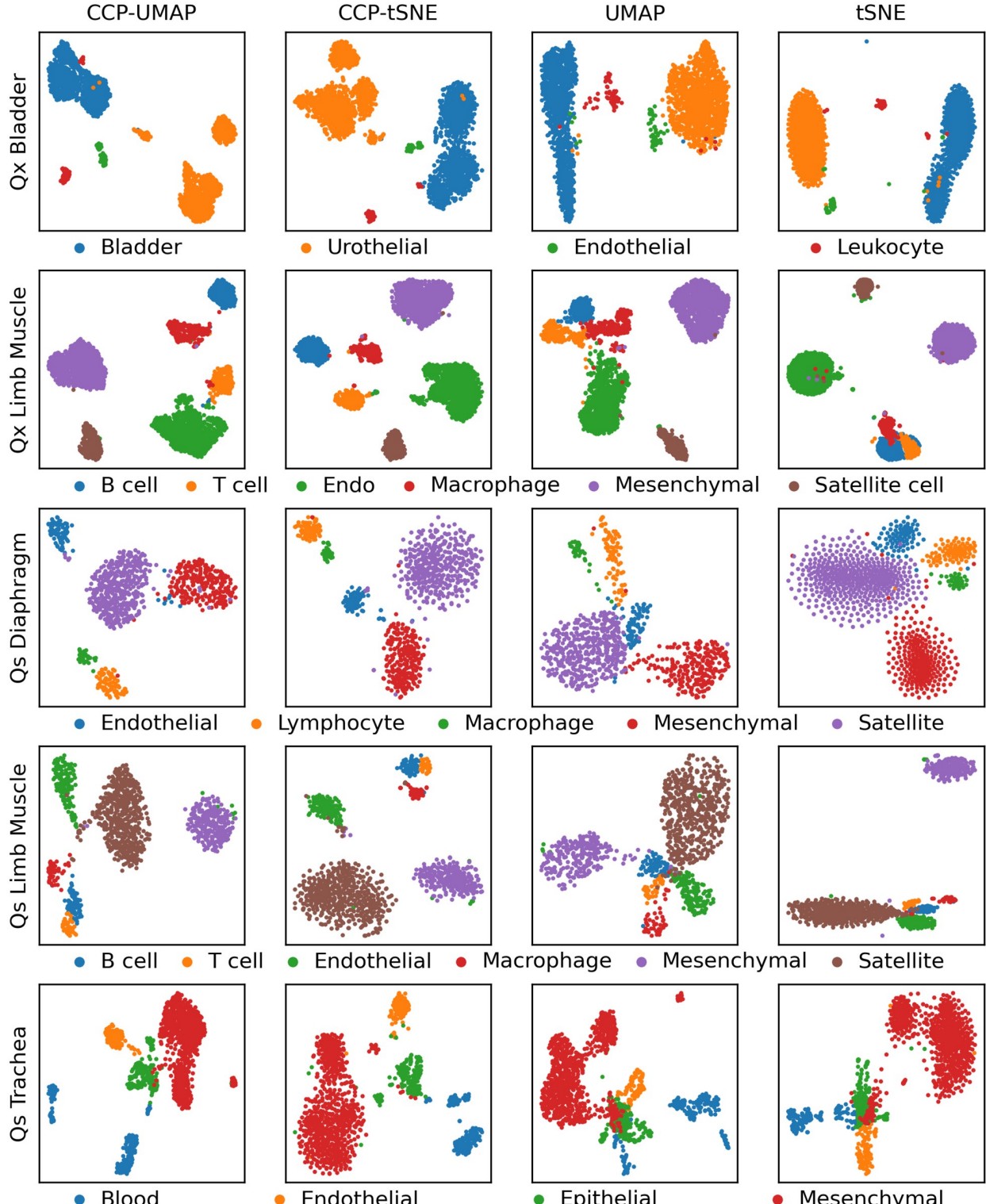

**Fig 3. Comparison of CCP-assisted UMAP and tSNE with standard UMAP and tSNE visualization on Quake dataset.** The rows correspond to Qx Bladder, Qx Limb Muscle, Qx Diaphragm, Qs Limb Muscle and Qs Trachea. Qx indicates scRNA-seq obtained used 10x genomic platform, and Qs indicate data obtained from SmartSeq2 platform. CCP was used to reduced the dimension to 300 super-genes. UMAP and tSNE were utilized to further reduce the dimension to 2 to obtain the visualization. Samples were colored according to the cell types provided by the original authors.

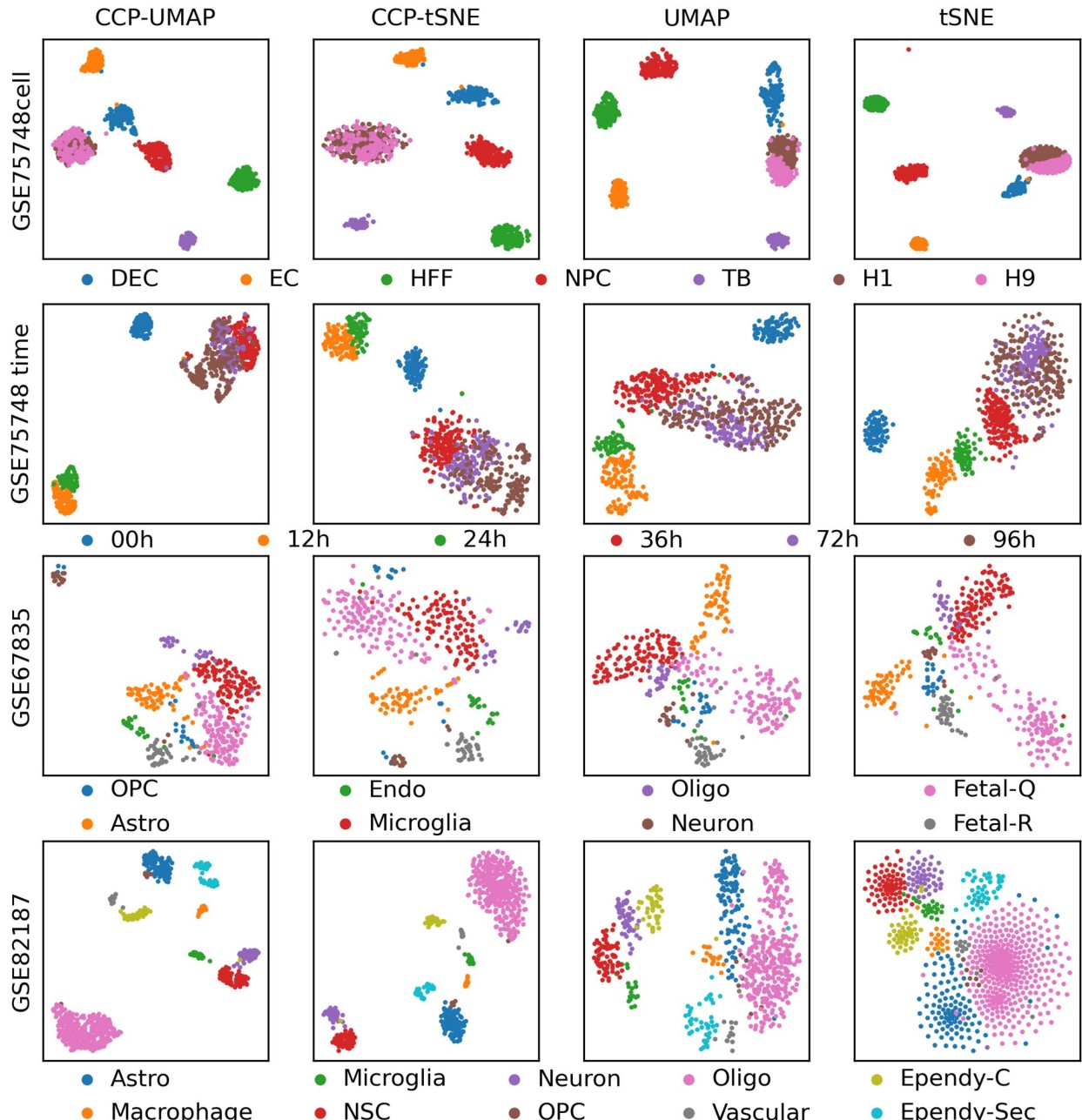

**Fig 4. Comparison of CCP-assisted UMAP and CCP-assisted tSNE with the standard UMAP and tSNE visualization on GSE75748 cell, GSE75748 time, GSE67835 and GSE82187.** CCP was used to reduced the dimension to 300 super-genes. UMAP and tSNE were utilized to further reduce the dimension to 2 to obtain the visualization. Samples were colored according to the cell types provided by the original authors.

In GSE75748 cell data, all the visualizations are similar. In [56], Chu obtained snapshots of lineage-specific progenitor cells that differentiated from H1 human embryonic stem (ES) cells and compared the gene profiles with undifferentiated H1 and H9 human ES cells as control. Most notably, H1 and H9 clustered together, which is consistent with our visualization. In GSE75748 time, all visualization is comparable. Chu et al [56] obtained snapshot of ES cell differentiation from pluripotency to definitive endoderm over the time period 0hr, 12hr, 24hr,

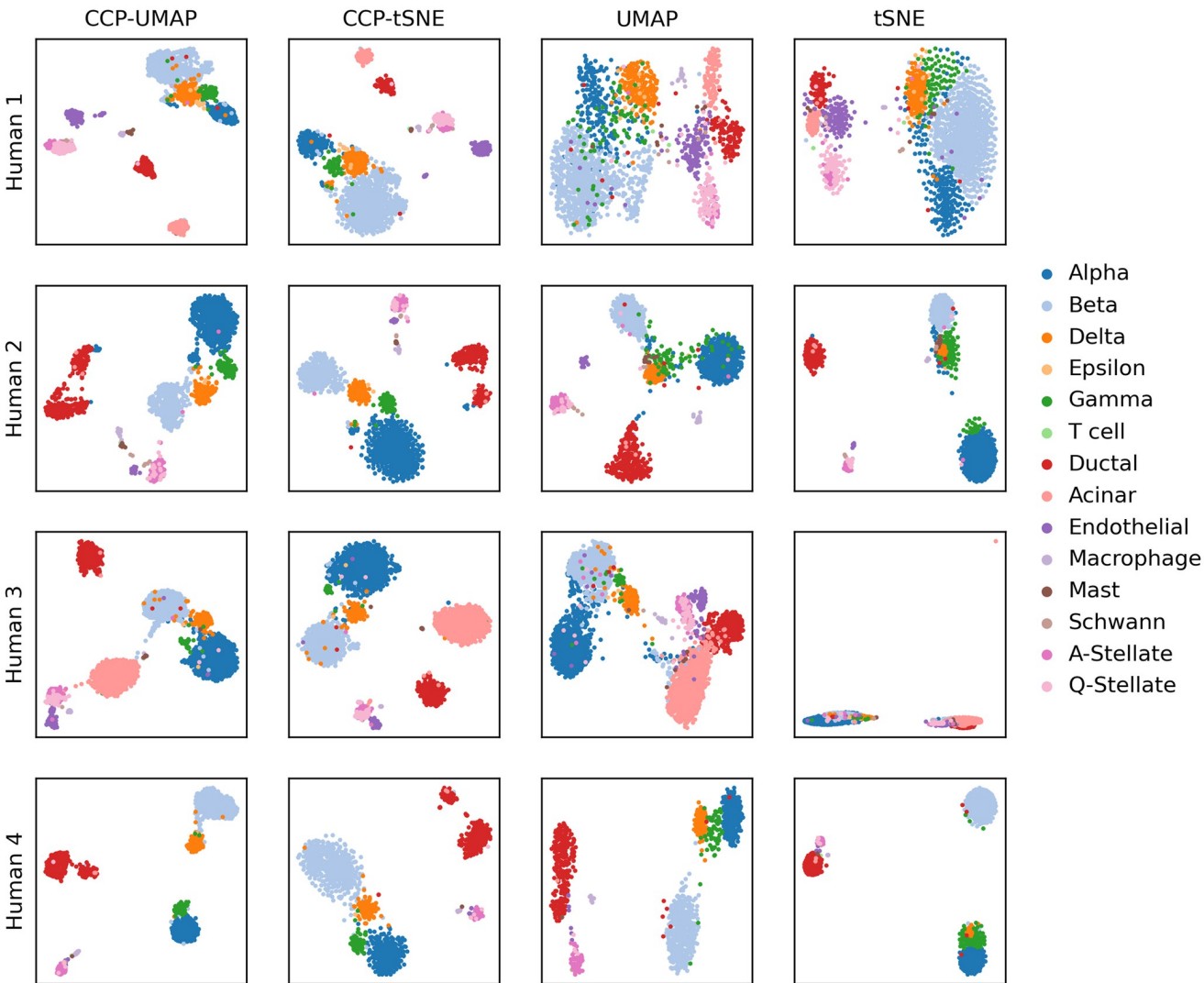

**Fig 5. Comparison of CCP-assisted UMAP and CCP-assisted tSNE with the standard UMAP and tSNE visualization on GSE84133 human dataset.**
Each row corresponds to 1 of the 4 patients. CCP was used to reduced the dimension to 300 super-genes. UMAP and tSNE were utilized to further reduce the dimension to 2 to obtain the visualization. Samples were colored according to the cell types provided by the original authors.

36hr, 72hr and 96hr. Chu noted the cells sequenced at 72hr and 96hr show relatively similar expression profiles, suggesting that the differentiation has completed by 72hr. We see from our visualization that the 72hr and 96hr cells form a cluster, 12hr and 24hr cells form a cluster, and 0hr cells form its own cluster, indicating that there is a clear distinction between the undifferentiated and the cells undergoing differentiation. In GSE67835, CCP-assisted visualization and its counter part have comparable result. Most notably, neurons cell from a distinct cluster in CCP-assisted visualization, whereas it does not in the standard visualization. In GSE82187 data, CCP-assisted UMAP and tSNE show a significant improvement over standard UMAP and tSNE visualization. Aside from astrocytes and OPC, all cell types form its own cluster. Standard UMAP and tSNE fail to show significant clustering of the different cell types.

Fig 5 show the comparison of CCP-assisted UMAP and tSNE with standard UMAP and tSNE visualization on Baron human dataset [55]. The rows correspond to the patients, and the

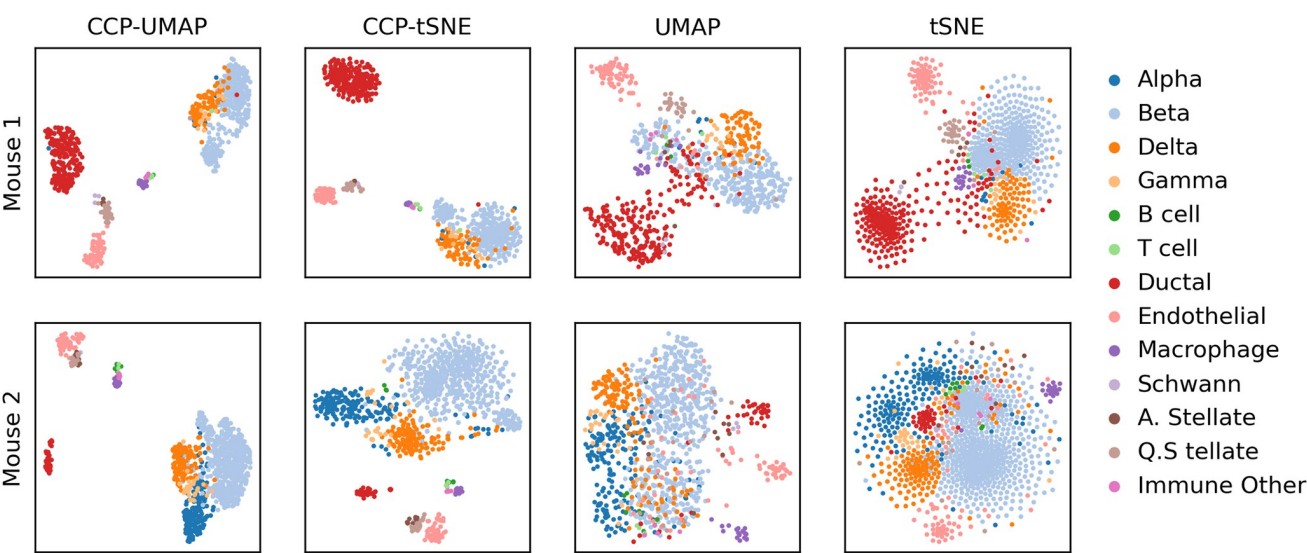

**Fig 6. Comparison of CCP-assisted UMAP and CCP-assisted tSNE with the standard UMAP and tSNE visualization on GSE84133 human dataset.**
The rows correspond to mouse 1 and 2. CCP was used to reduced the dimension to 300 super-genes. UMAP and tSNE were utilized to further reduce the dimension to 2 to obtain the visualization. Samples were colored according to the cell types provided by the original authors.

columns correspond to CCP-assisted UMAP, CCP-assisted tSNE, standard UMAP and standard tSNE visualization. The samples were colored according to the true cell type.

Overall, CCP-assisted visualizations show stronger clustering. In standard UMAP and tSNE visualizations across all patients, we noticed superclusters with unclear boundaries. Conversely, CCP-assisted visualizations display well-defined boundaries between cell types. Most notably is the clear differentiation of quiescent stellate (Q-Stellate) cells, alpha cells, and ductal cells across all patients, which is a distinction that isn't as evident in the standard visualizations. Additionally, standard tSNE visualization of patient 3 show instability in the standard tSNE algorithm, where the visualization do not differentiate the cell types.

Fig 6 show the comparison of CCP-assisted UMAP and tSNE with standard UMAP and tSNE visualization on Baron mouse dataset [55]. The rows correspond to the patients, and the columns correspond to mouse 1 and 2, and the columns correspond to CCP-assisted UMAP, CCP-assisted tSNE, standard UMAP and standard tSNE visualization. The samples were colored according to the true cell type.

CCP-assisted visualizations demonstrate significantly stronger clustering for both mouse samples. In the standard visualizations, beta cells are scattered among other cell types. Furthermore, in the data from mouse 2, alpha cells do not form a distinct cluster. Conversely, CCP-assisted visualizations distinctly cluster all cell types. Regarding mouse 1, the CCP-assisted visualization does not form a cluster for gamma cells, potentially due to the limited number of available gamma cells.

Fig 7 show the visualization of Murano, Romanov and Qs Lung data. The columns correspond to CCP-assisted UMAP, CCP-assisted tSNE, standard UMAP and standard tSNE visualization. The samples were colored according to the true cell type.

In the Muraro dataset, CCP-assisted UMAP exhibits a clear separation of A cells, D cells, B cells, and ductal cells. In contrast, the standard UMAP visualization presents these cells as a supercluster. The standard tSNE visualization indicates the instability of tSNE algorithm, where the visualization is unclear and dominated by and A cell outlier. Regarding the

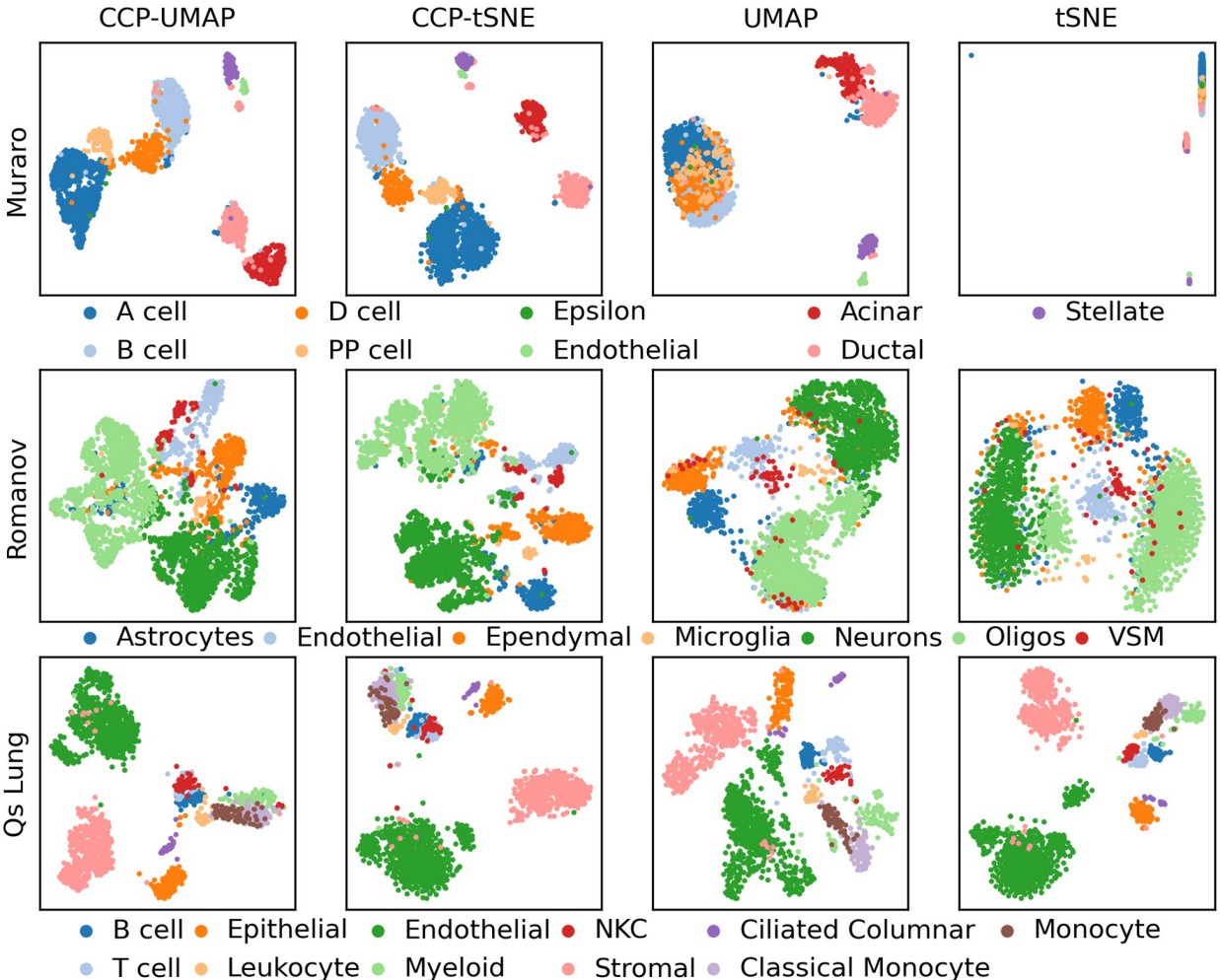

**Fig 7. Comparison of CCP-assisted UMAP and CCP-assisted tSNE with the standard UMAP and tSNE visualizationn on Muraro, Romanov and Qs Lung dataset.** CCP was used to reduced the dimension to 300 super-genes. UMAP and tSNE were utilized to further reduce the dimension to 2 to obtain the visualization. Samples were colored according to the cell types provided by the original authors.

Romanov dataset, all visualizations are relatively similar. CCP-assisted UMAP reveals a distinct cluster of astrocytes and ependymal cells, whereas both the standard UMAP and tSNE display a supercluster of these two cell types. Additionally, CCP-assisted UMAP and tSNE suggest two subclusters of VSM and endothelial cells, which are not discernible in the standard visualization. In the Qs Lung dataset, CCP-assisted and standard visualizations yield comparable results. While the standard tSNE separates monocytes from classical monocytes, CCP-assisted UMAP and tSNE portray a homogeneous clustering of these two cell types.

## Accuracy

To assess CCP's effectiveness as a primary dimensionality reduction tool for UMAP and tSNE, we conducted clustering using the Leiden algorithm within scanpy. We employed the adjusted Rand index (ARI) and normalized mutual information (NMI) to gauge accuracy by comparing the clustering results with the labels provided by the dataset's authors. It's important to note that these metrics do not measure absolute accuracy due to the absence of a gold standard

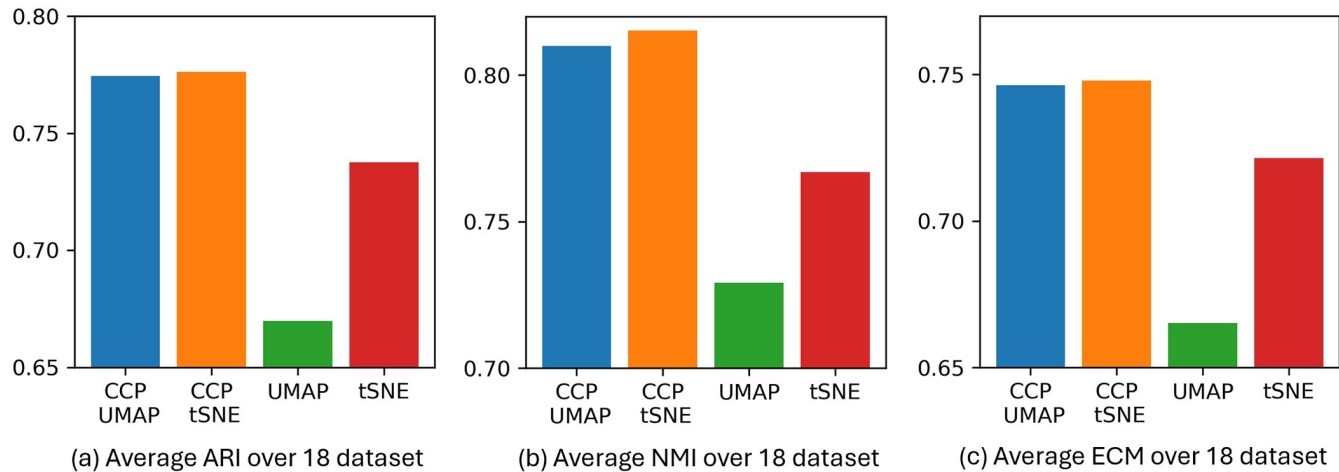

**Fig 8. The average ARI, NMI, ECM of 18 datasets.** 10 random initialization was used to compute CCP, CCP-assisted UMAP, CCP-assisted tSNE, standard UMAP and standard tSNE for each dataset. Leiden clustering was used to obtain the clustering results.

dataset for scRNA-seq. Additionally, we used the Element-Centric measure (ECM) [62] to evaluate cluster stability.

Fig 8 show the average ARI, NMI and ECM of CCP-assisted UMAP, CCP-assisted tSNE, UMAP and tSNE across 18 dataset. For each dataset, we conducted 10 random seeds to perform dimensionality reduction, utilizing Leiden clustering to generate clustering labels. These labels were then compared to the annotated cell types provided by the original authors.

CCP-assisted UMAP demonstrates a 22% improvement in ARI, 14% in NMI, and 15% in ECM over standard UMAP. Similarly, CCP-assisted tSNE improves standard tSNE by 11% ARI, 9% NMI and 8% in ECM. Additionally, CCP-assisted UMAP and tSNE have a higher ECM score, indicating that their clustering is more stable. Notably, both CCP-assisted UMAP and tSNE yield higher ECM scores, indicating more stable clustering. Interestingly, standard tSNE outperforms UMAP. However, UMAP's performance heavily relies on accurately finding nearest neighbors, which can be challenging with noisy, sparse, and high-dimensional data. CCP effectively reduces dimensions, enabling UMAP to find neighbors more effectively and resulting in improved visualization.

For a detailed comparison between CCP-assisted, PCA-assisted, and NMF-assisted visualizations, please refer to Section S2.3 in the S1 File. Additionally, we provide the ARI, NMI and ECM for each dataset in S2–S4 Tables of the S1 File.

## Discussion

### Large data

While CCP proves to be an efficient dimensionality reduction technique for datasets with a large number of features, such as in the case of scRNA-seq data, it may encounter limitations due to the necessity of computing cell-cell correlations for each super-gene. To address this challenge, for larger datasets, we propose a subsampling approach.

Let $\mathcal{Z} = \{\mathbf{z}_1, ..., \mathbf{z}_M\}$ be the training data used to develop a CCP model, and $\mathcal{Y} = \{\mathbf{y}_1, ..., \mathbf{y}_T\}$ be a new dataset or additional data. Using the training data, gene partitions $S^n$, cutoff distance $r_c^{S^n}$ and the connectivity $\eta^{S^n}$ are determined. Then, we embed new data to the

trained model, utilizing the following modification to Eq 5

$$x_i^n = \sum_{m=1}^{M} \Phi(\|\mathbf{y}_i^{S^n} - \mathbf{z}_m^{S^n}\|; \eta^{S^n}, \tau, \kappa), \qquad (6)$$

to obtain appropriate super-genes.

We verified the subsampling approach on GSE84133 human and Qx Spleen data. We combined all four patient's sequencing data into one superset for this analysis. We randomly subsampled 500, 1000, 1500, 2000, 2500, 3000 samples as a training data, and performed the subsampling under 10 random seeds. We projected the testing data using Eq 6, followed by Leiden clustering. ARI and NMI were computed, and the average scores are reported in Fig 9. Notably, both the GSE84133 human and Qx Spleen datasets exhibited consistent and stable results under varying number of subsampling.

Additionally, we also show that CCP-assisted UMAP and tSNE for both data when subsampling was utilized. Notably, all visualizations were comparable, underscoring the stability of CCP-assisted visualizations even under subsampling. For the computation time, subsampling scheme using 1000 samples took 152 seconds and 160 seconds on GSE84133 human and Qx Spleen data, respectively. Additional comparison can be found in the S1 File.

## Low variance genes

We have utilized LV-genes to enhance the predictive power of super-genes with a LV gene cluster. By using a high cutoff ratio, we can reduce the number of genes used in the feature partition, potentially resulting in a lower number of super-genes. To assess the impact of the cutoff ratio on the number of super-genes used for UMAP and tSNE visualizations, we conducted tests using GSE82187 and GSE75748 cell data. The discussion for GSE75748 cell data can be found in Section S3.2 of the S1 File.

Fig 10 show the effect of varying the number of super-genes and the cutoff ratio on the predictive power and visualization of GSE82187 data. We utilized 10 random seeds to generate CCP super-genes using different numbers of super-genes and cutoff ratios. Subsequently, Leiden clustering was applied to obtain cluster labels, and the ARI was computed utilizing the cell labels provided by the original authors. Notably, across all cutoff ratios, the ARI increases with an augmented number of super-genes, plateauing at a comparable level around 300 super-genes. This indicates the robustness of LV-gene.

Fig 10(c) show the visualization of CCP-assisted UMAP and tSNE at various cutoff ratio. For the visualization, 300 super-genes were utilized, and UMAP and tSNE was applied to the super-genes to reduce the dimension to 2. Samples were then colored according to the cell types provided by the original authors. Note that all the visualization are comparable, indicating the robustness of LV-gene under different cutoff ratio.

## Conclusion

CCP is a nonlinear data-domain dimensionality reduction technique that leverages gene-gene correlations to partition genes, and utilizes cell-cell correlation to generate super-genes. Unlike methods that involve matrix diagonalization, CCP can be directly applied as a primary dimensionality reduction tool to complement traditional visualization techniques like UMAP and tSNE. In our experiments with 18 datasets, CCP-assisted UMAP and CCP-assisted tSNE visualizations consistently outperformed the original UMAP and tSNE. On average, CCP-assised UMAP improves the standard UMAP visualization by 22% in ARI, 14% in NMI and 15% in ECM, and CCP-assisted tSNE improves standard tSNE by 11% ARI, 9% NMI and 8%

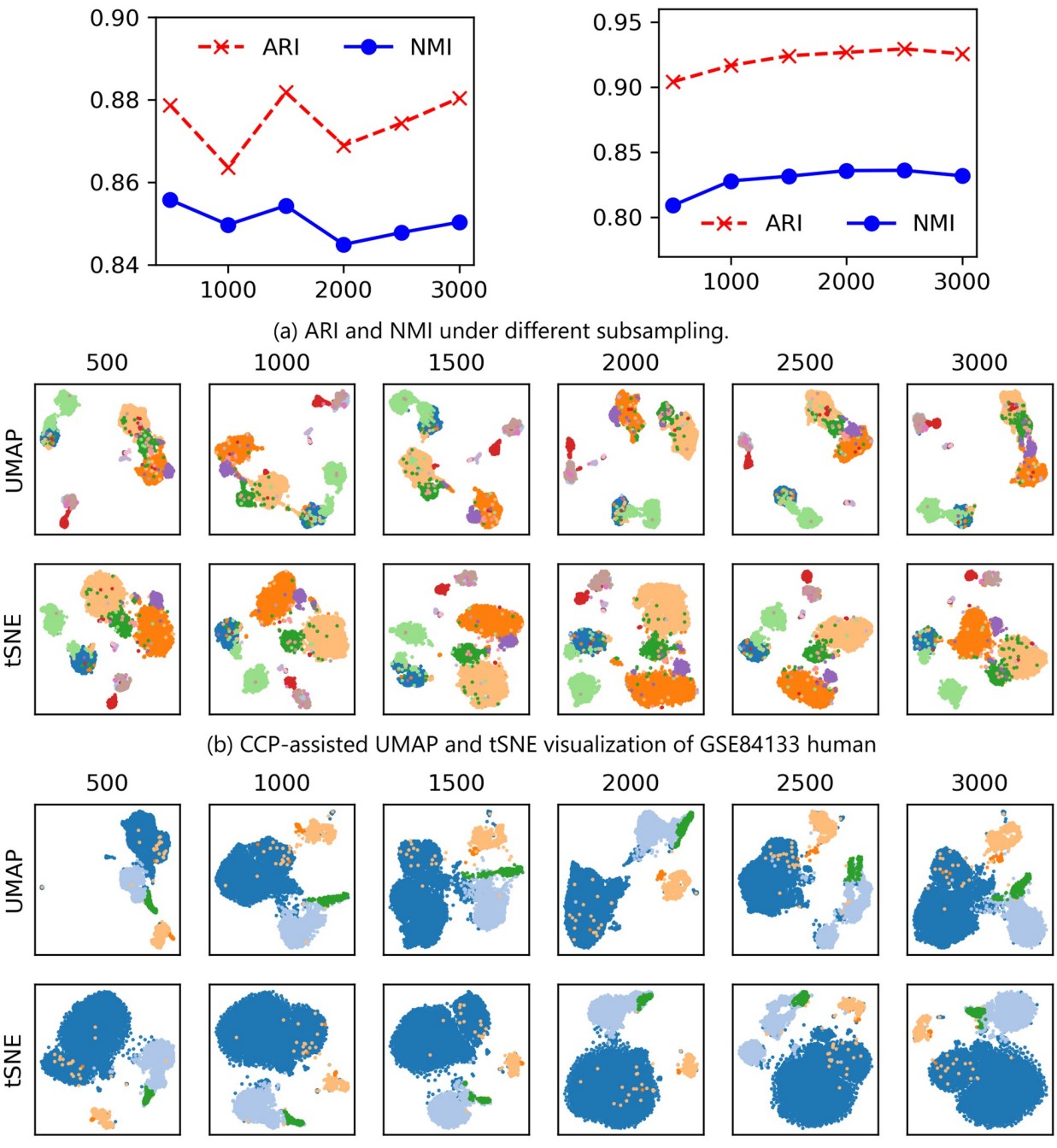

**Fig 9. UMAP and tSNE visualization of GSE84133 human and Qx Spleen data under different number of subsampling.** 300 super-genes were generated from CCP, and Leiden clustering was used to obtain the clustering results. (a) ARI and NMI under different subsampling values. Left figure shows the ARI and NMI for GSE84133 Human, where the 4 patient data were combined. Right figure shows the ARI and NMI of Qx Speen data. (b) CCP-assisted UMAP and tSNE of GSE84133 Human data under different number of subsampling. (c) CCP-assisted UMAP and tSNE of Qx Spleen data under different number of subsampling.

in ECM. Although the improvement for tSNE visualization is less than the improvement in UMAP, tSNE is sensitive to potential outliers and noise, where the visualization can become uninterpretable. CCP-assisted tSNE consistently show clear visualization in the 21 dataset we

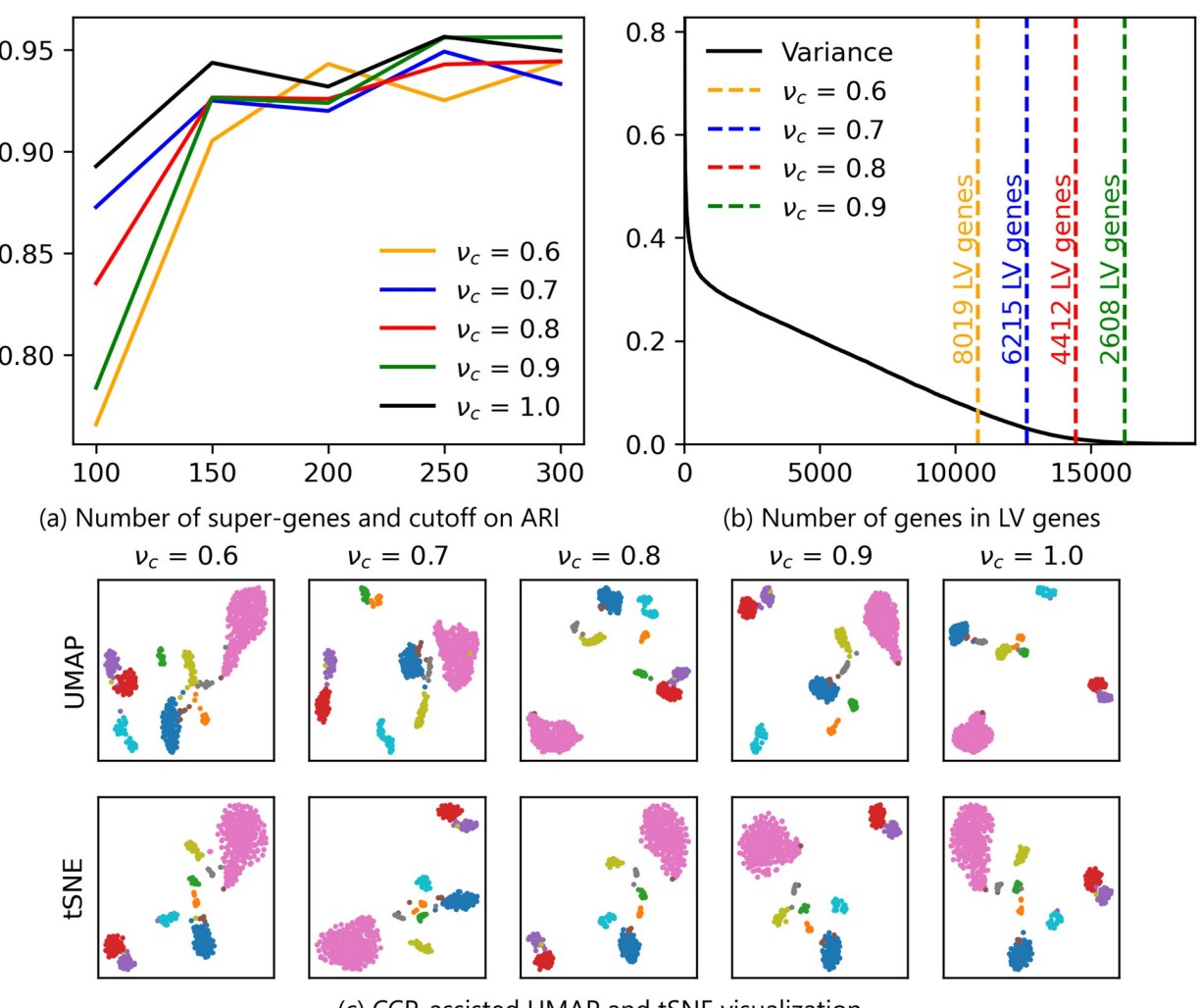

**Fig 10. Analysis of varying the cutoff ratio $v_c$ on clustering and visualization of GSE82187 data.** (a) ARI of leiden clustering when the number of super-genes and cutoff ratio is changed. (b) The number of genes in the LV-gene when $v_c$ is changed. (c) Top and bottom row shows the CCP-assisted UMAP and tSNE visualization, and the columns corresponds to $v_c$ = 0.6, 0.7, 0.8, 0.9. 300 super-genes were used to initialize UMAP and tSNE, and the samples were colored according to the true cell type.

have tested. Additionally, CCP-assisted visualization improves PCA-assisted and NMF-assisted visualization in the 21 dataset we have tested. However, CCP comes with some disadvantageous. For data with no clear gene-gene correlation, CCP will most likely not perform well. Additionally, although utilizing gene clustering removes the complication with computing distance in high dimensions, when the number of samples becomes large, the cell-cell correlation computation becomes time consuming. We show that subsampling via a training set is an effective approach to enable CCP for dealing with large data. One possible extension for gene clustering is to incorporate prior information, such as using known genes or utilizing known gene regulatory pathways, to guide in the clustering. Additionally, CCP can also be employed in many other single cell contexts, such as spatial transcriptomics and cell-cell communication, and for initializing deep learning methods.

## Supporting information

**S1 File. Supporting materials for analyzing scRNA-seq data by CCP-assisted UMAP and t-SNE.**
(PDF)

## Author Contributions

**Conceptualization:** Guo-Wei Wei.

**Data curation:** Yuta Hozumi.

**Formal analysis:** Yuta Hozumi.

**Funding acquisition:** Guo-Wei Wei.

**Investigation:** Yuta Hozumi.

**Methodology:** Yuta Hozumi.

**Project administration:** Guo-Wei Wei.

**Resources:** Guo-Wei Wei.

**Software:** Yuta Hozumi.

**Supervision:** Guo-Wei Wei.

**Validation:** Yuta Hozumi.

**Visualization:** Yuta Hozumi.

**Writing – original draft:** Yuta Hozumi, Guo-Wei Wei.

**Writing – review & editing:** Yuta Hozumi, Guo-Wei Wei.

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
