## [Decision Letter · Decision Letter 0]

13 May 2024

PONE-D-24-06894Analyzing scRNA-seq data by CCP-assisted UMAP and tSNEPLOS ONE

Dear Dr. Wei,

Thank you for submitting your manuscript to PLOS ONE. After careful consideration, we feel that it has merit but does not fully meet PLOS ONE’s publication criteria as it currently stands. Therefore, we invite you to submit a revised version of the manuscript that addresses the points raised during the review process.

The reviewers’ comments pointed out that a significant revision is required to improve and strengthen the current version of the manuscript. A better literature review should be incorporated into the manuscript. The mathematical formalism used in the manuscript needs to be carefully revised. The bibliography should be carefully revised and unified. A flowchart should be added to the manuscript to better explain the whole process. Larger datasets should be tested to show the scalability and general performance of the proposed approach. Please refer to the reviewers’ reports and the Reviewer’s Responses to Questions section for detailed comments, which could help you improve your manuscript. Please carefully address (and reply to) all the comments raised by all reviewers (this is mandatory).

We look forward to receiving your revised manuscript.

Kind regards,

Andrea Tangherloni

Academic Editor

PLOS ONE

Journal Requirements:

3. Thank you for stating the following financial disclosure: "NIH grants R01GM126189, R01AI164266, and R35GM148196

NSF grants DMS-2052983, DMS-1761320, and IIS-1900473

NASA grant 80NSSC21M0023

MSU Foundation

Bristol-Myers Squibb 65109

Pfizer"

4. Please expand the acronym “NIH, NSF, NASA and MSU” (as indicated in your financial disclosure) so that it states the name of your funders in full.

5. Thank you for stating the following in the Acknowledgments Section of your manuscript: "This work was supported in part by NIH grants R01GM126189, R01AI164266, and R35GM148196 , NSF grants DMS-2052983, DMS-1761320, and IIS-1900473, NASA grant 80NSSC21M0023, MSU Foundation, Bristol-Myers Squibb 65109, and Pfizer."

Please remove any funding-related text from the manuscript and let us know how you would like to update your Funding Statement. Currently, your Funding Statement reads as follows: "NIH grants R01GM126189, R01AI164266, and R35GM148196

NSF grants DMS-2052983, DMS-1761320, and IIS-1900473

NASA grant 80NSSC21M0023

MSU Foundation

Bristol-Myers Squibb 65109

Pfizer"

6. Please remove your figures from within your manuscript file, leaving only the individual TIFF/EPS image files, uploaded separately. These will be automatically included in the reviewers’ PDF.

Additional Editor Comments :

The reviewers’ comments pointed out that a significant revision is required to improve and strengthen the current version of the manuscript. A better literature review should be incorporated into the manuscript. The mathematical formalism used in the manuscript needs to be carefully revised. The bibliography should be carefully revised and unified. A flowchart should be added to the manuscript to better explain the whole process. Larger datasets should be tested to show the scalability and general performance of the proposed approach. Please refer to the reviewers’ reports and the Reviewer’s Responses to Questions section for detailed comments, which could help you improve your manuscript. Please carefully address (and reply to) all the comments raised by all reviewers (this is mandatory).

Reviewers' comments:

Reviewer's Responses to Questions

**Comments to the Author**

1. Is the manuscript technically sound, and do the data support the conclusions?

Reviewer #1: Yes

Reviewer #2: Yes

Reviewer #3: Yes

Reviewer #4: Yes

Reviewer #5: Yes

Reviewer #6: Yes

2. Has the statistical analysis been performed appropriately and rigorously? 

Reviewer #1: Yes

Reviewer #2: Yes

Reviewer #3: Yes

Reviewer #4: I Don't Know

Reviewer #5: Yes

Reviewer #6: No

3. Have the authors made all data underlying the findings in their manuscript fully available?

Reviewer #1: Yes

Reviewer #2: Yes

Reviewer #3: Yes

Reviewer #4: Yes

Reviewer #5: Yes

Reviewer #6: Yes

4. Is the manuscript presented in an intelligible fashion and written in standard English?

Reviewer #1: Yes

Reviewer #2: Yes

Reviewer #3: Yes

Reviewer #4: Yes

Reviewer #5: Yes

Reviewer #6: Yes

5. Review Comments to the Author

Reviewer #1: Authors utilize the proposed method CCP as an initialization tool for uniform manifold approximation and projection (UMAP) and t-distributed stochastic neighbor embedding (tSNE) and reported enhancement of CCP assisted UMAP and tSNE over UMAP and tSNE. The paper is well written, organised and I believe the propose method may be good alternative for scRNA-seq data visulaization. I have some suggestions and comments givn below:

1. Nowadays, many deep learning-based dimensionality reduction methods are introduced and they show promising results. It is not clear what are the advantages of the proposed methods over those deep learning-based methods apart from CCP has the ability to work on small size datsets. Some comparison results of CCP assisted UMAP and tSNE, along with PCA and NMF and deep learning assisted UMAP and tSNE may be reported.

2. Nowadays, scRNA-seq datasets contain millions of cells. Authors have considered datasets size less than 10000 cells. I suggest to incorporate results on some large datasets.

3. Authors have reported some impressive results. However, apart from average NMI and ARI reported in Fig 7, detailed results on each of the methods in terms of ARI, NMI on each datasets is expected in tabular format may be in supplementary.

4. In scRNA-seq data analysis running time is also another issue. It would be helpful for reader if authors can report a comparison of the proposed method in terms of running time and memory usage. I also suggest to incorporate systems specification on which the method was tested.

5. The result reported in Fig 8 can be extended by considering a datset of much larger size.

6. I wanted access to the method online. But the link is not working.

7. Typos: line number 236

8. Need captions in supporting figures in supplementary.

Reviewer #2: The authors proposed to utiliza correlated clustering and projection to analyze scRNA-seq data. The work may have been a laborious one that I should appreciate the authors' time and patient to come up with some results. However, there are several problems that deduct from the quality of this manuscript. Below are several comments on this work.

1. The article does not reflect the time consumption performance of this method. Is it suitable for large-scale data sets (Data of more than 10,000 cells)? Are there any disadvantages compared to standard UMAP or t-SNE?

2. Is it scientific that the Urothelial in the Qx Bladder data set in Figure 2 is divided into three sub-clusters? Is it biologically significant?

3. Is it scientific for Mesenchymal in the Qx Trachea data set in Figure 2 to cluster two sub-clusters into one category?

4. Ductal in the Human4 data set in Figure 4 is divided into two clusters in CCP-assisted. Is this biologically meaningful?

Reviewer #3: 1.English expressions need to be edited more careful and more native, in this manuscript, there are some mistakes. For example, “The The samples were colored according to the true cell type” in line 205 on page 13.

2. I suggest the authors should add a flowchart in the manuscript to show the process very well.

3. Important computational models about dimensionality reduction and feature selection in scRNA-seq analysis should be cited. Some recommended studies are helpful (PMIDs: 36642414, 36924730, and 37660567).

4. The authors should carefully check and unify the information of references. Some references lack the information of volume or contain the wrong page number.

Reviewer #4: Correlated Clustering and Projection (CCP) is a data dimensionality reduction method previously designed by the authors. During the dimensionality reduction process, CCP separately processed the low variance (LV) genes.

In this manuscript, the authors found that the reduced data obtained from CCP dimensionality reduction can improve the visualization effect of UMAP and tSNE. From the authors’ experiments, the effect is good.

Major concerns:

1. CCP first clusters genes, which may mask the importance of different genes. Some clusters may be sets of low expression genes, while some others may be sets of high expression genes. Some marker genes are originally unique, but they may be in the same cluster as other non-marker genes, or there may be many marker genes in a certain cluster, or some clusters may not have any marker genes. That is, during the gene clustering process, the uniqueness of marker genes may be reduced? Please reply if there are any possibilities that I mentioned here.

2. It is best to draw a flowchart or an example diagram for the method. Solely using descriptions and formulas can confuse us on subtle differences, such as whether x_m ^ n in line 105 and x_i ^ n in formula (5) represent the same thing.

3. The authors set the number of super-genes to 300 for all datasets, but the number of genes in the original datasets used in the paper is different. Why not dynamically adjust the number of super-genes based on the number of genes?

4. P5, line 118, What weights are these w_ {im} and how to set?

Minor comments:

P4 line 104, Does z_m ^ {S ^ n} represent the expression vector of the corresponding cluster?

Typos: Page10 Line 167, dataa data

Reviewer #5: 1. Provide a more detailed explanation of how CCP partitions genes and identifies super-genes to enhance understanding.

2. Include specific details about CCP implementation and parameter settings for reproducibility.

3. Compare CCP with other dimensionality reduction techniques such as PCA or autoencoders to contextualize its performance.

4. Detail the statistical tests used to claim improvements in UMAP and tSNE, including p-values, to substantiate your results.

5. Describe the 18 datasets used, including their sources and characteristics, to assess the generalizability of the findings.

6. Discuss how CCP specifically handles the sparsity in scRNA-seq data, a critical challenge in the field.

7. Explain the biological relevance of the improvements made by CCP to highlight its impact on the field.

8. Provide information about any software available for implementing CCP to aid in adoption and further research.

9. Identify limitations of CCP and suggest areas for future research to guide subsequent studies.

10. Improve the quality of graphical representations of UMAP and tSNE results to better communicate your findings.

11. Incorporate recent literature to position CCP within the current research landscape and ensure the study is up-to-date.

Advancing single-cell RNA-seq data analysis through the fusion of multi-layer perceptron and graph neural network.

iDNA-OpenPrompt: OpenPrompt learning model for identifying DNA methylation

Reviewer #6: In this manuscript, the authors propose the usage of the Correlated Clustering and Projection (CCP) method as an initiation step for t-SNE and UMAP. They have presented results which show substantial improvement over ordinary t-SNE and UMAP when CCP is used as a preprocessing step. However, there are a few concerns.

Major concerns:

1. The CCP algorithm used in this work cannot be considered as a benchmark since it is not yet published as a peer-reviewed journal or as a part of any conference proceedings.

2. The authors have not mentioned what values of parameters they have used for t-SNE and UMAP. As it is quite well known that these parameters (for e.g. perplexity in t-SNE or n_neighbors and min-dist in UMAP) are very sensitive, probably fine-tuning them could have achieved the same result, which is not clear.

Infact, the codebase https://github.com/hozumiyu/CCP-scRNAseq-UMAP-TSNE/blob/main/main.py reveals the usage of one pair of values for the dataset GSE57249 and another pair of values for others. No particular guideline for setting these values have been provided.

3. Again, based on the tutorial notebook provided in github, it is not clear why a z-score normalization is required before running UMAP, even when the dataset is preprocessed using CCP. The authors have not presented any discussion about it in the manuscript.

4. The comparative results shown in this manuscript are mainly qualitative. For quantitative comparison, the authors have used only three measures, namely ARI, NMI and ECM. However, they have reported an average over 18 datasets (Figures 7 and S7). This is not an acceptable result at all. Should we assume that results for the 18 datasets have a large variation?

Also, how to access performance in situations where ground truth labels are not available? In addition to ARI, NMI, ECM, pairwise distance preservation before and after CCP-UMAP or CCP-tSNE could be measured.

5. In section 4, the authors mention "Large" datasets. However, the largest dataset used in this study does not have more than 10k cells. Considering technological advancements in the field of scRNA-seq, a million cells is now a reality. So, questions remain as to what is a really large amount? The authors should give an estimate of the running time requirement for the subsample-based strategy for such a dataset.

Minor:

--------

1. ECM - Please change "Element-centeric" to "Element-Centric"

2. Page 16 - $ instead of %

3. "Section ??" - Section reference missing in S1.2 in supporting material

4. Page 8 - S1.3 - $ instead of %

6. PLOS authors have the option to publish the peer review history of their article (what does this mean?). If published, this will include your full peer review and any attached files.

Reviewer #1: No

Reviewer #2: No

Reviewer #3: No

Reviewer #4: No

Reviewer #5: No

Reviewer #6: **Yes: **Rajat K. de

---

## [Author Response · Author response to Decision Letter 0]

7 Aug 2024

Comment From Editor

Comment 1: Please ensure that your manuscript meets PLOS ONE’s style requirements, including those for file naming.

Answer 1: We have followed the submission formatting guidelines

Comment 2: Please note that PLOS ONE has specific guidelines on code sharing for submissions in which author-generated code underpins the findings in the manuscript. In these cases, all authorgenerated code must be made available without restrictions upon publication of the work.

Answer 2: All the code used to generate our results is submitted to Github https://github.com/hozumiyu/CCPscRNAseq-UMAP-TSNE

Comment 3: Thank you for stating the following financial disclosure. Please state what role the funders took in the study.

Answer 3: The funders had no role in study design, data collection and analysis, decision to publish, or preparation of the manuscript. We have indicated this in the cover letter.

Comment 4: Please expand the acronym “NIH, NSF, NASA and MSU” (as indicated in your financial disclosure) so that it states the name of your funders in full. This information should be included in your cover letter; we will change the online submission form on your behalf.

Answer 4: NIH: National Institute of Health, NSF: National Science Foundation, NASA: National Aeronautics and Space Administration, MSU: Michigan State University

Comment 5: Thank you for stating the following in the Acknowledgments Section of your manuscript:”This work was supported in part by NIH grants R01GM126189, R01AI164266, and R35GM148196, NSF grants DMS-2052983, DMS-1761320, and IIS-1900473, NASA grant 80NSSC21M0023, MSU Foundation, Bristol-Myers Squibb 65109, and Pfizer.” We note that you have provided funding information that is not currently declared in your Funding Statement. However, funding information should not appear in the Acknowledgments section or other areas of your manuscript. We will only publish funding information present in the Funding Statement section of the online submission form. Please include your amended statements within your cover letter; we will change the online submission form on your behalf.

Answer 5: We have removed the acknowledgement section in our manuscript. The current funding statement is appropriate, and no further addition is neccessary.

Comment 6: Please remove your figures from within your manuscript file, leaving only the individual TIFF/EPS image files, uploaded separately. These will be automatically included in the reviewers’ PDF.

Answer 6: We have removed the figures from the manuscript.

------------

Reviewer 1

Comments: Authors utilize the proposed method CCP as an initialization tool for uniform manifold approximation and projection (UMAP) and t-distributed stochastic neighbor embedding (tSNE) and reported enhancement of CCP assisted UMAP and tSNE over UMAP and tSNE. The paper is well written, organised and I believe the propose method may be good alternative for scRNA-seq data visulaization. I have some suggestions and comments givn below:

Answer: We appreciate the reviewer for the work.

Comment 1: Nowadays, many deep learning-based dimensionality reduction methods are introduced and they show promising results. It is not clear what are the advantages of the proposed methods over those deep learning-based methods apart from CCP has the ability to work on small

size datsets. Some comparison results of CCP assisted UMAP and tSNE, along with PCA and NMF and deep learning assisted UMAP and tSNE may be reported.

Answer 1: Although deep learning models have become prevalent in scRNA-seq analysis, they may give very little insight to what are happened within the deep learning models. CCP, similar to NMF, performs the reduction purely in the data domain, and also preserves the nonnegativity of the original data. Such method is much more interpretable than deep-learning models. Additionally, our visualization approach is targeted for single-experiment reduction, whereas deep learning models often need careful curation of multiple data sources. Our approach has been compared with UMPA and tSNE, showing the improvement.

Comment 2:Nowadays, scRNA-seq datasets contain millions of cells. Authors have considered datasets size less than 10000 cells. I suggest to incorporate results on some large datasets.

Answer 2: The goal of the subsampling procedure was to tackle larger data. We have added additional data SCP1749 from the Broad Institute’s single cell repository, which consists of more than 10,000 cells. The result was added to the supporting materials. We are developing a GPU version of CCP which enables us to deal with millions of cells (however, this is out of the scope of the present paper).

Comment 3:Authors have reported some impressive results. However, apart from average NMI and ARI reported in Fig 7, detailed results on each of the methods in terms of ARI, NMI on each datasets is expected in tabular format may be in supplementary.

Answer 3: We have added the results for each data as a table in the supporting materials section S1.3.1.

Comment 4: In scRNA-seq data analysis running time is also another issue. It would be helpful for reader if authors can report a comparison of the proposed method in terms of running time and memory usage. I also suggest to incorporate systems specification on which the method was tested.

Answer 4: We benchmarked our method using Michigan State University’s high performance computing cluster, which utilized AMD EPYC 7H12 Processor. For all the test, we used 4 CPU core with 16gb of memory. We have added the description into the manuscript. As mentioned, with a GPU version, the benchmark will change.

Comment 5: The result reported in Fig 8 can be extended by considering a datset of much larger size.

Answer 5: We have added additional data SCP1749 from the Broad Institute’s single cell repository, which consists of more than 10,000 cells. The result was added to the supporting materials.

Comment 6: I wanted access to the method online. But the link is not working.

Answer 6: The code to replicate our results can be found at https://github.com/hozumiyu/CCPscRNAseq-UMAP-TSNE

Comment 7: Typos: line number 236

Answer 7: We have fixed the typo on line 236 14$ to 14%

Comment 8: Need captions in supporting figures in supplementary.

Answer: We have added captions to all the figures in the supporting materials

----------

Reviewer 2

Comments: The authors proposed to utilize correlated clustering and projection to analyze scRNA-seq data. The work may have been a laborious one that I should appreciate the authors’ time and patient to come up with some results. However, there are several problems that deduct from the quality of this manuscript. Below are several comments on this work.

Answer: We appreciate the reviewer for the review of our work.

Comment 1: The article does not reflect the time consumption performance of this method. Is it suitable for large-scale data sets (Data of more than 10,000 cells)? Are there any disadvantages compared to standard UMAP or t-SNE?

Answer 1: Because the method requires distance computation for each super-genes, the time complexity is proportional to the number of super-genes and the distance calculation. For larger data, using a subset of the data to train CCP will reduce the computational time, and an analysis is performed in the discussion. Standard UMAP and tSNE require distance calculation, which is difficult for high dimensional data (i.e., curse of dimensionality). CCP overcomes this limitation by reducing the original dimension to 300 super-gene. Moreover, the distance calculation is much more reliable using the CCP, resulting in a much better result. We are developing GPU and multiple CPU based CCP methods, which will enable us to deal with millions of cells.

Comment 2: Is it scientific that the Urothelial in the Qx Bladder data set in Figure 2 is divided into three sub-clusters? Is it biologically significant?

Answer 2: This is an interesting question. The real cause may be multiple. For example, the data may have contaminated. There may be batch effect. One needs to carefully investigate the original data collection protocol before assuming a biological significance, which is beyond the scope of our work.

Comment 3: Is it scientific for Mesenchymal in the Qx Trachea data set in Figure 2 to cluster two sub-clusters into one category?

Answer 3: Currently, no method is able to correctly classify all single cells, due to its limitation or noisy experimental data. This is reason that the research community is continuously developing new methods.

Comment 4: Ductal in the Human4 data set in Figure 4 is divided into two clusters in CCPassisted. Is this biologically meaningful?

Answer 4: As stated early, the real cause may be multiple. For example, the data may have contaminated. There may be batch effect. One needs to carefully investigate the original data collection protocol before assuming a biological significance, which is beyond the scope of our work

----------

Reviewer 3

Comments 1: English expressions need to be edited more careful and more native, in this manuscript, there are some mistakes. For example, “The The samples were colored according to the true cell type” in line 205 on page 13.

Answer 1: We have gone through additional rounds of editing to fix these mistakes.

Comments 2: I suggest the authors should add a flowchart in the manuscript to show the process very well.

Answer 3: We have added a flowchart.

Comments 3: Important computational models about dimensionality reduction and feature selection in scRNA-seq analysis should be cited. Some recommended studies are helpful (PMIDs: 36642414, 36924730, and 37660567).

Answer: We appreciate additional reference, and have incorporated them into the introduction.

Comments 4: The authors should carefully check and unify the information of references. Some references lack the information of volume or contain the wrong page number.

Answer4: We have gone through each reference to ensure that the references are correct and made the reference more consistent.

----------

Reviewer 4

Comments: Correlated Clustering and Projection (CCP) is a data dimensionality reduction method previously designed by the authors. During the dimensionality reduction process, CCP separately processed the low variance (LV) genes. In this manuscript, the authors found that the reduced data obtained from CCP dimensionality reduction can improve the visualization effect of UMAP and tSNE. From the authors’ experiments, the effect is good.

Answer: We thank the reviewer for the positive feedback.

Comments 1: CCP first clusters genes, which may mask the importance of different genes. Some clusters may be sets of low expression genes, while some others may be sets of high expression genes. Some marker genes are originally unique, but they may be in the same cluster as other non-marker genes, or there may be many marker genes in a certain cluster, or some clusters may not have any marker genes. That is, during the gene clustering process, the uniqueness of marker genes may be reduced? Please reply if there are any possibilities that I mentioned here.

Answer 1: It is certainly possible for the marker genes to be clustered with non-marker genes as long as these genes exhibit similar pattern as the marker genes. We would like to emphasize that CCP is entirely unsupervised. We have not incorporated prior knowledge for the super-gene, but we would like to consider guided feature partition using prior knowledge in the future.

Comments 2: It is best to draw a flowchart or an example diagram for the method. Solely using descriptions and formulas can confuse us on subtle differences, such as whether xn m in line 105 and xni in formula (5) represent the same thing.

Answer 2: We have added a flowchart to better clarify our methodology.

Comments 3: The authors set the number of super-genes to 300 for all datasets, but the number of genes in the original datasets used in the paper is different. Why not dynamically adjust the number of super-genes based on the number of genes?

Answer 3: We fixed the number of super-genes to be 300 to be entirely parameter-free in our analysis. Additionally, for potential use in the deep learning data preprocessing, a fixed number of output is needed. However, it would be up to the user to adjust the number of super-gene as well as the LV-gene for the particular dataset.

Comments 4: P5, line 118, What weights are these wim and how to set?

Answer 4: We have set wim = 1 in this work. We would like to further explore this parameter to incorporate other information, such as spatial information in spatial transcriptomics.

Comments 5:P4 line 104, Does zSn m represent the expression vector of the corresponding cluster?

Answer 5: z_m^Sn represents the super-gene n for the m-th cell.

Comments 6: Page10 Line 167, dataa –¿data

Answer 6: We have fixed the typo.

------------

Reviewer 5

Comments 1: Provide a more detailed explanation of how CCP partitions genes and identifies super-genes to enhance understanding.

Answer 1: We have added more details to clarify the methodology. Additionally, we added a flowchart to clarify the method.

Comments 2: Include specific details about CCP implementation and parameter settings for reproducibility.

Answer 2: We have used the same parameters for all the tests. We set κ = 2, τ = 6 and n = 300. We have added the choice of parameter in the manuscript. We have also added these explanation in the github repository.

Comments 3: Compare CCP with other dimensionality reduction techniques such as PCA or autoencoders to contextualize its performance.

Answer 3: The comparison with PCA and NMF has been carried out and can be found in the supporting materials.

Comments 4: Detail the statistical tests used to claim improvements in UMAP and tSNE, including p-values, to substantiate your results.

Answer 4: We have added additional table in the supporting materials that show the ARI, NMI and ECM of each data we have tested.

Comments 5: Describe the 18 datasets used, including their sources and characteristics, to assess the generalizability of the findings.

Answer 5 In the supporting materials, we have a table characterizing the dataset, which include the host species, and statistics on the expression profiles.

Comments 6: Discuss how CCP specifically handles the sparsity in scRNA-seq data, a critical challenge in the field.

Answer 6: Sparsity is indeed a challenge that every method faces when working with scRNA-seq data. In our method, we utilize LV-gene to collect low-varying genes to obtain a stronger signal. Additionally, CCP partitions genes according to their similarity (i.e., similar sparsity patterns in sparse genes).

Comments 7: Explain the biological relevance of the improvements made by CCP to highlight its impact on the field. 

Answer 7: Gene expression has many correlations. For example, genes belonging to the same pathway may exhibit similar expression patterns. CCP identifies these patterns in an unsupervised fashion (i.e., does not need prior knowledge). Since no prior knowledge is utilized, our method provides additional perspective in scRNA-seq analysis in an unbiased fashion.

Comments 8: Provide information about any software available for implementing CCP to aid in adoption and further research.

Answer 8: Code and data availability is described Sec. 6.

Comments 9: Identify limitations of CCP and suggest areas for future research to guide subsequent studies.

Answer 9: The limitation of CCP is in the computational time. We have proposed subsampling to circumvent this limitation, and discussed the stability under subsampling in the Discussion section. In the future, we would like to extend the work to spatial data. In particular, the weights wim can incorporate prior information, such as spatial information of the cell. Additionally, the current CCP is entirely unsupervised. We would like to incorporate marker genes as a prior to guide in the gene-partitioning procedure in the future. We have added these comments in the manuscript. Finally, we are working on GPU and parallel versions of CCP.

Comments 10: Improve the quality of graphical represent

---

## [Decision Letter · Decision Letter 1]

25 Sep 2024

Analyzing scRNA-seq data by CCP-assisted UMAP and tSNE

PONE-D-24-06894R1

Dear Dr. Wei,

We’re pleased to inform you that your manuscript has been judged scientifically suitable for publication and will be formally accepted for publication once it meets all outstanding technical requirements.

Kind regards,

Andrea Tangherloni

Academic Editor

PLOS ONE

Comments from PLOS Editorial Office:

We note that one or more reviewers has recommended that you cite specific previously published works in an earlier round of revision. As always, we recommend that you please review and evaluate the requested works to determine whether they are relevant and should be cited. It is not a requirement to cite these works and you may remove them before the manuscript proceeds to publication. We appreciate your attention to this request.

Reviewers' comments:

Reviewer's Responses to Questions

**Comments to the Author**

1. If the authors have adequately addressed your comments raised in a previous round of review and you feel that this manuscript is now acceptable for publication, you may indicate that here to bypass the “Comments to the Author” section, enter your conflict of interest statement in the “Confidential to Editor” section, and submit your "Accept" recommendation.

Reviewer #1: All comments have been addressed

Reviewer #3: (No Response)

Reviewer #5: All comments have been addressed

2. Is the manuscript technically sound, and do the data support the conclusions?

Reviewer #1: Yes

Reviewer #3: (No Response)

Reviewer #5: Yes

3. Has the statistical analysis been performed appropriately and rigorously? 

Reviewer #1: Yes

Reviewer #3: (No Response)

Reviewer #5: Yes

4. Have the authors made all data underlying the findings in their manuscript fully available?

Reviewer #1: Yes

Reviewer #3: (No Response)

Reviewer #5: Yes

5. Is the manuscript presented in an intelligible fashion and written in standard English?

Reviewer #1: Yes

Reviewer #3: (No Response)

Reviewer #5: Yes

6. Review Comments to the Author

Reviewer #1: The authors have addressed all my comments and I believe quality of the paper improved a lot. I don't have any further comments.

Reviewer #3: (No Response)

Reviewer #5: All comments have been thoroughly addressed, and my recommendations have been accepted and implemented.

7. PLOS authors have the option to publish the peer review history of their article (what does this mean?). If published, this will include your full peer review and any attached files.

Reviewer #1: **Yes: **Hussain Ahmed Chowdhury

Reviewer #3: No

Reviewer #5: No

---

## [Editor Report · Acceptance letter]

1 Oct 2024

PONE-D-24-06894R1 

PLOS ONE

Dear Dr. Wei, 

I'm pleased to inform you that your manuscript has been deemed suitable for publication in PLOS ONE. Congratulations! Your manuscript is now being handed over to our production team.

Kind regards, 

on behalf of

Dr. Andrea Tangherloni 

Academic Editor

PLOS ONE